# Towards successful business process improvement – An extension of change acceleration process model

**Maha Syed Ibrahim**[1]*, **Aamer Hanif**[2], **Faheem Qaisar Jamal**[3], **Ali Ahsan**[4]

**1** Department of Business and Engineering Management, Sir Syed CASE Institute of Technology, Islamabad, Pakistan, **2** School of Management (AUSOM), Air University, Islamabad, Pakistan, **3** Department of Engineering Management, NUST College of E&ME, Islamabad, Pakistan, **4** Faculty of Management Sciences, Foundation University Islamabad, Islamabad, Pakistan

* maha.syed@gmail.com

**Data Availability Statement:** All relevant data are within the paper and its Supporting Information files.

## Abstract

Change Acceleration Process model (CAP) emerged in early 90's as a set of principles for accelerating change management efforts in organizations. Business Process Improvement (BPI) projects open avenues of opportunity and success for organizations in this highly competitive era. However, most of these projects fail due to lack of commitment, communication, scope creep and inadequate resources. This research attempts to study industry relevant factors most critical to success of a BPI Project in the highly competitive telecom sector. Modified Delphi technique employing a panel of telecom professionals was adopted in order to determine the critical success factors (CSFs) after a thorough review of the literature. Exploratory factor analysis was performed to map extracted factors to the five agents of change. Research outcome defines the relevant CSFs in terms of vision, skills, incentives, resources and action plan. A significant contribution of this research is an extended CAP model for implementation of BPI projects. Practical implications of this research are utilization of the proposed model for BPI project success.

## Introduction

Business Process Management (BPM) is the science that ensures consistent outcomes and the practice to seize improvement opportunities by overseeing performance of cross-functional work in organizations [1]. This highly challenging era demands organizations to improve continuously just to stay competitive in the run. It is a perennial responsibility of the management to analyze their business processes and improve them to be more efficient and productive. Complex nature of the business environment demands rapid and significant changes. To survive in such environments, managers are compelled to respond to these changes swiftly by revising their business processes. The identification and improvement of business processes comes under the umbrella of BPM. Telecommunication industry is a rapidly growing industry worldwide and faces many challenges. This requires these organizations to be more responsive to change [2]. Advancement in management sciences have brought various tools and

**Funding:** The author(s) received no specific funding for this work.

**Competing interests:** The authors have declared that no competing interests exist.

techniques which help organizations to be more receptive and adaptive e.g. Business Process Improvement (BPI) lets organizations to improve their business processes gradually and continuously [3]. Successful implementation of BPI interventions is really challenging, as a BPI project demands attention from various business functions making it an expensive proposition. It has been seen that 60–70% of the BPI projects fail and are not completed [4, 5]. The key problems identified behind failure of BPI projects are the lack of acknowledgment of the risks that potentially confront organizations for successful implementation. The identification of risks lie in the people, process and product paradigm [6]. Specifically for the services sector, the problem has some additional dimensions like the quality of service, service response time and service performance enhancement [7, 8].

Generally, implementation of BPI projects has been focusing primarily on process design and system configurations while the areas involving soft factors and intangibles related to employees and customers have not been extensively explored in literature [9]. Moreover, the impact of culture, motivation, and people side of BPI need further exploration specifically with respect to the services sector [10]. A few models have been brought forward that consider culture and role of leadership as the pillars for successful implementation, however there is no known model for BPI implementation focusing on BPI implementation success dictated by project champions and people management [10, 11, 12, 13].

## Pakistani telecom industry

In recent years, telecom companies across the globe have faced difficulties and challenges for Pakistan have been no different. With more than 150 million users, the industry employs about 1.36 million persons in the five major players in the market. Although the consumption of mobile data has increased overall revenues, cash flows have declined. This is mainly attributed to the fact that telecom companies have made heavy investments for their wireless 3G, 4G and 5G networks. Pakistan will continue to face challenges in the fiscal year 2019–20 as the companies will participate in the competition for market share in a market that has reached its saturation. The consumers have now become more aware and now the need has arisen for the companies to continuously improve their business processes and innovate in order to exceed the customer expectations.

## Change accelerated process analysis

In 2001, General Electric came up with a practical and less complex model named as Change Accelerated Process (CAP). CAP is a widely used change management tool successfully used for the implementation of change in many organizations. The CAP model (Fig 1) illustrates the elements that are common to all successful change initiatives as an organization moves from its current state, through the transition, and to the improved or future state. The model is presented in Fig 1.

The objective of this research is to bridge the identified gap by extension of General Electric's CAP for implementation of BPI Projects in telecommunication industry of Pakistan. The proposed extension of CAP identifies industry specific CSF's for BPI, enabling the practitioners to make informed decisions that actually yield results, thus making a significant addition to existing literature.

## Research contribution and novelty

This paper proposes a comprehensive framework for BPI implementation including the key elements of change encompassing the critical success factors. In doing so, the paper addresses a clear gap in literature that calls for a comprehensive framework to assist the BPI

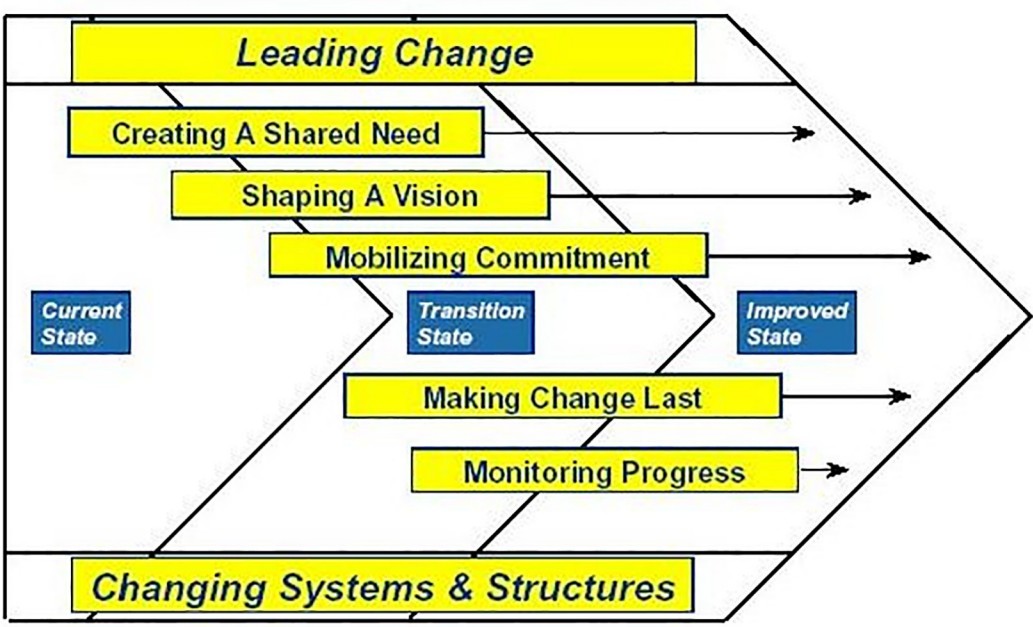

**Fig 1. Change accelerated process [14].**

implementation for achieving desired results. The main contribution of this paper is an extension of the CAP model for successful implementation of BPI projects in the telecom sector. Some other related contributions of this research are:

1. Identification of industry specific CSFs.

2. Definition of CSFs in terms of the 5 key agents of change namely: vision, skill, incentives, resources, and action plan

3. Research in the area of BPI, and the factors critical to its implementation success in context of Pakistan.

The expected outcome of this research is an extension of CAP model for BPI project implementation, introducing a fresh perspective for implementation.

## Literature review

Over the last couple of decades, implementation of BPI projects has been studied, through both theoretical and practical lenses. Various studies have identified Critical Success Factors (CSFs) as well as critical failure factors (CFFs) of successful implementation of a BPI project [15]. This provided a rich basis from which the researchers can get deeper understanding of the contributing factors. The authors have attempted to summarize the CSFs found in the literature. The CSFs have evolved since the conception of BPI due to global competition, rapidly changing business environment and developing technologies. This evolution has played a role in fine-tuning the CSFs of BPI projects. The refinement of literature over the years is described below:

One of the earliest studies suggests that BPI initiatives are successful if they are aligned with organizational strategy [16]. Literature subsequently focused on the Business Process Management (BPM). It identified CSFs that revolved around project actors, BPM teams, organizational leaders/ leadership, communication, commitment and politics [17, 18, 19, 20].

As BPI matured, CSFs were categorized into organization, process and technology specific factors [21, 22, 23, 24]. Different studies resulted into addition of multiple CSFs like learning (amongst employees), organizational culture (that is more apt and adaptable to change) [6, 25, 26, 27], resource management [28] and a more structured approach [29].

When customer needs claimed much attention, some papers also referred to it as a CSF [30, 31] With the increasing dependency on technology, BPI and business process redesign (BPR) tools were identified as CSFs [32, 33, 34]. Environment has emerged as a CSF in recent studies [27, 35, 36]. The organization dimension for BPI success has also been explored. Study of effectiveness of a BPI framework for a particular industry or organization and its applicability to another organization is also a recent research trend [37]. Keeping in mind the organizational dimension, a comprehensive list of CSFs that have been found reliable in literature are presented in S1 Appendix.

With a threshold of at least four citations (during the exhaustive review of the literature) for a success factor, the authors have formulated a list of 22 CSFs. Although the CSFs were validated in previous researches, further reliability of correct formulation of the list was ensured by showing it to more than 25 field experts of telecom industry and then through Delphi technique. Since CAP is widely used in the industry as a change management tool and has proven results [14], the authors have proposed an extension of CAP for BPI project implementation. Another viewpoint for change management exists in form of Knoster's model which identifies five key elements required to govern the process of change [38]. These five elements are shown in Fig 2, which also depicts the result of missing any one of these elements. For example, missing vision will result into confusion, lack of skills will cause anxiety, lack of incentives will result in gradual change, lack of resource will cause frustrations and absence of action plan will result into false starts.

The need for a more robust and people driven approach for BPI projects implementation is time and again established in studies [8, 11]. Recently identified top ten reasons for process improvement project failures include "lack of commitment and support from top management; poor communication practices; incompetent team; inadequate training and learning; faulty selection of process improvement methodology and its associated tools/techniques; inappropriate rewards and recognition system/culture; scope creepiness; sub-optimal team size and composition; inconsistent monitoring and control; and resistance to change" [39]. Although success factors have been identified in many researches, there is a need for a framework to facilitate successful implementation of BPI projects. The authors have recognized this gap and have made use of the elements of change as the foundations of this research. The authors have analyzed and established a link between the industry relevant CSFs and the elements of change.

## Materials and methods

The ethical considerations are stated prior to proceeding with the research design.

### Ethical considerations

The autonomy of individual respondents for this research was given due consideration by the researchers and all participation in the survey was voluntary. Confidentiality of participants and informed consent were specifically ensured. All participants were informed that their identity and individual responses were to be treated as anonymous and utilized only for the purpose of this research.

This is a mixed method study and is conducted in two phases. First phase is qualitative in nature while the second phase is quantitative in nature. The element of bias from the qualitative part (Delphi technique) was addressed by maintaining control over the process and by following a structured approach involving judgment of a number of experts from the field.

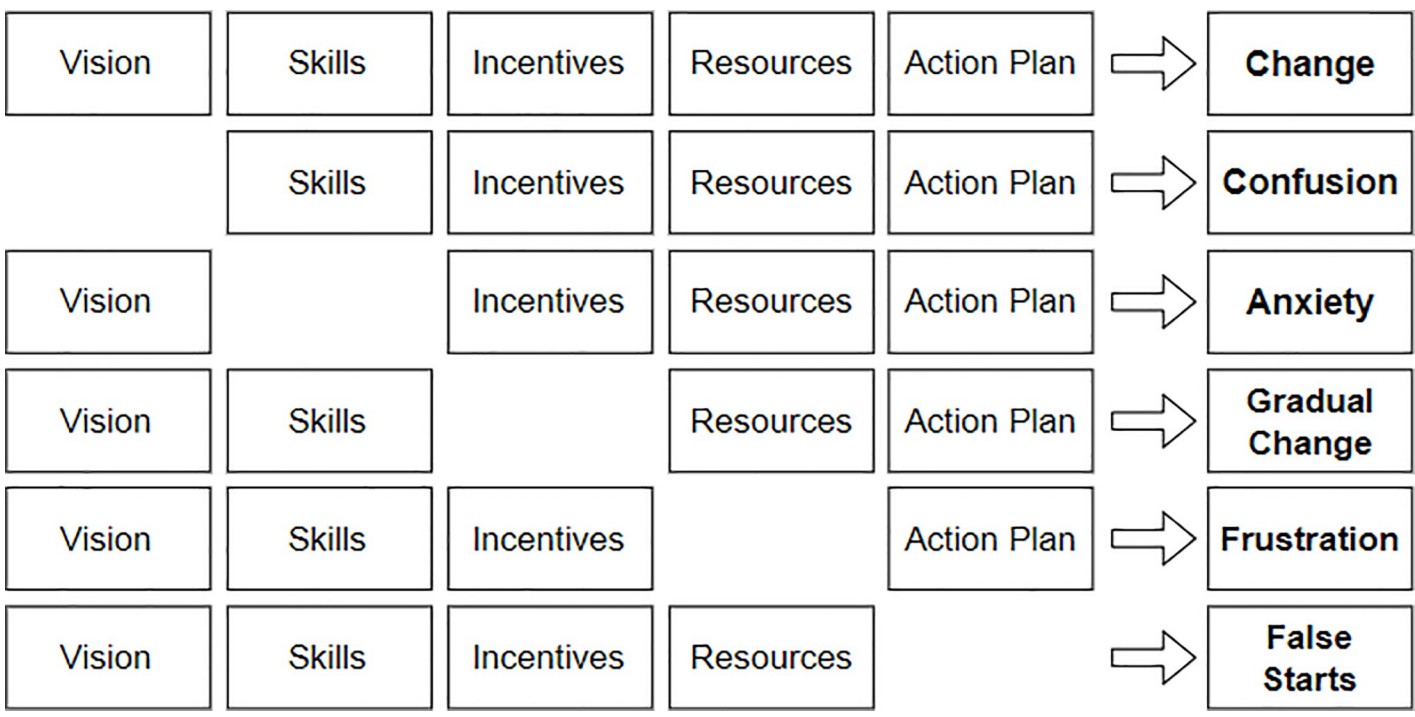

**Fig 2. Effects of missing elements [38].**

### Phase 1: The Delphi technique

Delphi technique has been used to finalize the CSFs for BPI in Telecom Industry of Pakistan. Delphi is used *"to explore or expose underlying assumptions or information leading to differing judgments; and to seek out information which may generate a consensus on the part of the respondent group;"*[40]. Delphi technique is used to gain clarity and arrive at a consensus on an area where there is extensive literature that makes the problem under discussion haphazard [41, 42]. Therefore, this technique is appropriate for our study since there is lack of consensus on the CSFs that are relevant to the telecom industry.

Delphi technique is used for gaining consensus from a panel of experts by undergoing multiple rounds where information is fed back to the panel after empirical analysis of the data from each round [43]. The cardinal aspects of Delphi are:

- Sampling and use of experts

- Anonymity of Delphi participants

- Iterations

- Controlled Feedback of responses

- Statistical aggregation of group responses

### Sampling and use of experts for Delphi

For Delphi technique, individuals with adequate knowledge of the topic under study are used as subjects and their opinions are requested to reach a consensus. In a study by McKenna, these subjects are referred to as a panel of informed individuals called *'experts'* [44]. The

selection of 'experts' is used as recommended by Keeney [41]. Delphi method is considered appropriate and used for BPI projects in organizations. Its structured and academically rigorous approach maintains controls over any expected bias [45]. For this study, the 'experts' chosen are Business Process Managers (BPMs) who have the experience of undertaking BPI projects for more than six years in the telecom industry of Pakistan. From this point forward, the experts are denoted as BPMs in this research.

The studies indicate that the sample size may vary from 4 to 3000 [46]. A recommendation has been found in the literature for usage of minimal number of subjects that would seek to verify the responses and would commit their attention throughout the Delphi process [42]. The authors selected a panel of 30 BPMs from the telecom industry for the application of the Delphi technique. Out of these only 26 responded in the first and 22 responded in subsequent rounds. Among the 26 who responded, there were 21 males and 5 females. All participants held masters degree and had an average work experience of at least seven years in the telecom industry. The sample size of 26 for this study is considered appropriate as similar studies with less sample sizes have been conducted and published [47].

## Questionnaire design for Delphi

In Delphi technique, a questionnaire is tweaked and modified in all rounds. A panel of experts who validated the process ensures content validity. Delphi concludes with a questionnaire which is then used for Exploratory Factor Analysis.

**Round 1.** Use of structured questionnaire, for the first round created after exhaustive study of the literature is more effective in driving the full potential of Delphi technique [42]. A questionnaire, with an exhaustive list of CSFs from the literature, was presented in Round 1. The BPMs were asked for their opinion on whether they considered these factors to be critical for the success of BPI Projects. They were also given the opportunity, if they would deem important to add any other CSF to the list.

**Round 2.** The BPMs were asked to rate the CSFs on a Likert scale from 1–5 (5 = most critical to success, and 1 = not critical to success). The Likert scale has been indicated as a standard way to analyze the relative importance of issues for Delphi iterations [48]. A 5-point Likert scale has been adopted as it has been used as an apt scale for deriving consensus on CSFs in various similar studies [38, 49, 50]

**Round 3.** The BPMs were provided with the panel mean rating and standard deviation for each CSF, and the BPMs were given the opportunity to review and revise the rating for each CSF

## Results of Delphi analysis

The results and analysis of subsequent rounds of Delphi technique are described below in a stepwise manner. Note that the results of Round 1 are used in Round 2 and similarly Round 2 results are used in Round 3. Detailed description of results and analysis is provided below:

**Round 1 CSF identification, results & analysis.** Modified Delphi technique was used for establishing round 1. Moreover, BPMs were also given the opportunity to add any CSF to the list provided. However, the respondents agreed that the CSFs presented to them were exhaustive and did not need any further addition. Therefore, none of the participants added any other CSF to the list.

The results collected from BPMs were analyzed. The agreement among 70–80% of the experts on a factor is generally considered as the selection criterion[43] while some researchers have opted for as low as 51% of agreement [40]. However, the BPMs consulted for this research

suggested that 80% agreement amongst the BPMs should be considered as the selection criterion.

The factors over which more than 80% of the BPMs had agreed upon being the most critical for success were then passed on to Round 2 for further iterations. Subsequently application of BPI toolbox (56%), project initiation and completion (0.76%), level of IT investment (0.64%), standardization of the process (0.76%), use of external support (0.44%), learning organizational culture (0.72%), were discarded as the CSFs, as the percentage of agreements is less than 80%.

**Round 2 Ranking, results & analysis.** The BPMs were asked to identify the relative importance of the CSFs for successful BPI on a Likert scale from 1 to 5 (1 = not at all critical to success, 2 = slightly critical, 3 = neutral, 4 = critical to success, 5 = very critical to success of BPI projects). Out of the 26 respondents that participated in Round 1, only 22 responded in Round 2. The data from these questionnaires was then analyzed in Statistical Package for Social Sciences (SPSS). The reliability statistic Cronbach's Alpha value of 0.779 (signifying the reliability [51] is also computed to measure the consistency of responses over successive rounds. The respondents were divided amongst two groups 'Leadership' and 'Management' to further analyze the results. This division into two groups was made on the basis of their designation in the organization. Out of the 22 respondents, 11 managers were classified as "management" and remaining 11 directors were classified as "leadership" category. The means of the responses were computed separately and arranged into descending order. The CSFs were "ranked" depending on the order of the means. Ranking was done to statistically analyze the consensus agreements [47].

Two statistical tests Spearman's rank correlation (rho) and Kendall's rank order correlation (tau-b) were calculated using SPSS for the two groups, 'Leadership' and 'Management'. The computed values of both 0.795 and 0.934 (for rho and tau-b respectively) exceeded the critical values. It was concluded that a statistical significant relationship existed amongst expert responses (at a significance level of 5%, 2-tailed) [51]. The value of 3.5 for overall mean was considered as a cut-off point [47]. Consequently, the top 13 CSFs with more than 3.5 overall mean score were selected for re-evaluation in Round 3.

**Round 3 Ranking, results & analysis.** The results of Round 2 helped in identifying the factors that were most critical to success of a BPI project. The reduced list of CSFs was again passed on to the 22 respondents for further reaching towards a group consensus. In Round 3 respondents were provided their previous ratings and mean and standard deviation (SD) of the group consensus was also provided. The panel was also given the provision to further evaluate and restate the rating on the same Likert scale for the CSFs. The means were again calculated and CSFs were arranged in descending order. The CSFs were also ranked according to their order. Ranking is done to statistically analyze the consensus agreements [47]. The responses of the 22 respondents depicted in Table 1 gives an overview of their opinions.

Same test procedures that were applied in Round 2 were applied again, as shown in Table 2, and it was found again that the values exceeded the critical values.

The consistency in results of the two rounds is calculated and compared as shown in Table 3.

## Concluding Delphi

The results of the two rounds were collated and the percentage improvement over the rounds is calculated as shown in Table 3. The results show the values of Kendall's coefficient had a 6% improvement and Spearman's rank correlation coefficient had an improvement of 1%. The results depict that there will be no substantial change in the results if the Delphi is iterated

**Table 1. Consensus measurement and ranks in round 3 (n = 22).**

| CSF Name | Overall | | Leadership | | Management | |
|---|---|---|---|---|---|---|
| | Mean | Rank | Mean | Rank | Mean | Rank |
| People change management | 5.000 | 1 | 5.000 | 1 | 5.000 | 1 |
| Understanding of the process | 4.925 | 2 | 5.000 | 2 | 4.850 | 4 |
| Involvement of organizations stake holders and leadership | 4.904 | 3 | 4.906 | 3 | 4.902 | 2 |
| Communication | 4.862 | 4 | 4.824 | 4 | 4.901 | 3 |
| Appointment of process owners | 4.765 | 5 | 4.730 | 5 | 4.800 | 5 |
| Resources allocation | 4.429 | 6 | 4.359 | 8 | 4.500 | 6 |
| Customer focus | 4.398 | 7 | 4.549 | 6 | 4.248 | 9 |
| Value realization | 4.395 | 8 | 4.445 | 7 | 4.345 | 7 |
| Performance measurement | 4.299 | 9 | 4.268 | 9 | 4.330 | 8 |
| Supporting organizational structure | 4.141 | 10 | 4.182 | 10 | 4.10 | 10 |
| People training and empowerment | 3.904 | 11 | 3.906 | 11 | 3.902 | 11 |
| Scope change management | 3.572 | 12 | 3.553 | 12 | 3.592 | 12 |
| Process improvement road map | 3.515 | 13 | 3.551 | 13 | 3.480 | 13 |

further. Since both the leadership and management have arrived at a consensus, we can safely move ahead with these 13 CSFs. With the help of Delphi analysis, we were able to narrow down from our list of 22 critical success factors to 13 factors that are considered most critical to success of a BPI project. The final list nominates the CSFs to be:

- Involvement of organizations stakeholders and leadership

- Performance measurement

- Supporting organizational structure

- People training and empowerment

- Appointment of process owners

- Communication

- Customer focus

- Understanding of the process

- Process Improvement road map

- People change management

- Value realization

- Scope change management

- Resources allocation

**Table 2. Consensus measurement for round3.**

| Sig (2- tailed); p = 0.05 | | Results |
|---|---|---|
| Kendall's tau_b Correlation Coefficient | 0.846** | 0.846>0.359 $H_0$ is Rejected |
| Spearman's rho Correlation Coefficient | 0.945** | 0.945>0.560 $H_0$ is Rejected |

**Table 3. Percentage improvement after round 2 and 3.**

| Kendall's rank-order Correlation Coefficient (tau-b) | | % Improvement |
|---|---|---|
| Round 2 | 0.795 | 6% |
| Round 3 | 0.846 | |
| **Spearman rank-order Correlation Coefficient (rho)** | | **% Improvement** |
| Round 2 | 0.934 | 1% |
| Round 3 | 0.945 | |
| **Reliability Statistics (Cronbach's (a))** | | **% Improvement** |
| Round 2 | 0.779 | 1% |
| Round 3 | 0.788 | |

## Phase 2: Exploratory factor analysis

Exploratory factor analysis (EFA) was conducted to consolidate the results of Delphi rounds. *EFA is the most appropriate technique when there is no prior hypothesis about factors or patterns of measured variables* [52]. In this case, the authors have applied EFA to the 13 CSFs to determine if the existence of any underlying relationships among CSFs. The four assumptions of EFA are: normal variables, linear relations, minimum correlation and sample size with a cases/ items ratio of at least 5:1 (for the 13 CSFs determined, it implies about 65 responses).

## Sample and sample size for EFA

The questionnaire finalized as the result of Delphi was used for EFA and floated to the BPI departments in the five major telecom organizations in Pakistan. The telecom organizations in Pakistan have central formal or informal BPI departments, whilst for nation-wide BPI projects, geographically distributed project managers are appointed as projects actors. The employees that are assigned the BPI projects are trained accordingly. Going forward we can notify them as the BPI teams for the sake of simplicity in this research. These BPI teams are the stratification characteristic of this population. This survey was administered to the employees that work on the BPI projects. The BPI teams are spread across the organization and contain 4–6 members. Proportionate stratified sampling technique was used to collect data. The BPI employees from 5 telecom companies are divided into 5 stratum corresponding to the population size of each strata, different clusters are assigned. Among the 29 clusters a total of 268 observations were deduced as the sample size.

## EFA questionnaire

Delphi technique resulted in a questionnaire that asked the respondents to rate each of the 13 CSFs from 1–5 on a Likert scale (5 = most critical, 4 = critical. 3 = neutral, 2 = slightly critical, 1 = Not critical at all). This questionnaire was floated among the designated sample.

## Results of exploratory factor analysis

The questionnaire formalized at the conclusion of Delphi technique was then floated within the designated clusters. Among the 268 designated respondents, 247 responded. Among the 247 who responded, there were 176 males and 71 females. Average work experience of the respondents was between 4 and 6 years. 45 respondents held a bachelors degree whereas 202 participants had masters degree. The reliability of the responses was checked by calculating Cronbach alpha which was 0.77 signifying the reliability of responses [53]. The data was then analyzed using exploratory factor analysis so that the factors that have similar contribution in

**Table 4. Factor loadings.**

| Comp# | Factor Name | Loadings |
|---|---|---|
| 1 | Understanding of the Process | 0.865 |
| | Realizing Value | 0.834 |
| | Communication | 0.856 |
| | Resource Allocation | 0.810 |
| | Inv. Of Organization Stakeholders and Leadership | 0.765 |
| 2 | Appointment of Process Owners | 0.802 |
| | People Change Management | 0.776 |
| 3 | Performance Measurement | 0.843 |
| 4 | Supporting Organizational Structure | 0.875 |
| | People Training and empowerment | 0.627 |
| 5 | Scope Change Management | 0.691 |
| | Process Improvement Road Map | 0.548 |
| | Customer Focus | -0.786 |

the model could be grouped together. This will give us the knowledge that how these CSFs of a BPI intervention participate for the successful implementation of the project. The exploratory factor analysis was performed on SPSS; Varimax (orthogonal) rotation technique with Kaiser Normalization was used [52].

It was observed that five extracted components explained 79.77% variability with the loss of information of less than 21%. The components thus derived had 41.05%, 14.46%, 11.08%, 8.15% of the total variance explained for components 1, 2, 3, 4 & 5, named as vision, skills, incentive, resources and action plan respectively. The loadings of the components are given in Table 4.

## Extension of CAP model

The Knoster model for managing complex change provides a framework with five basic elements to facilitate change management. This research makes an effort to integrate the elements of that model with the CAP model for BPI project implementation. The five elements of change and their proposed mapping to CAP model are given below:

1. "Vision" mapped to "shaping vision"

2. "Skills" mapped to "creating a shared need"

3. "Incentives" mapped to "monitoring progress"

4. "Resources" mapped to "making change last"

5. "Action Plan" mapped to "mobilizing commitment"

Fig 2 clearly explains that change is only possible when the five elements of change are present. The five key components derived from EFA were mapped on the agents of change. The conceptual basis of this mapping is described in Table 5.

Business process improvement is a continuous process allowing organizations to improve gradually. There are numerous BPI projects being carried out in an organization, focusing on reducing costs, delays and redundancies. It has been established that the implementation of these initiatives is a challenging task [54]. The research has answered this by providing an extension of CAP for BPI success. It was found that only two, one, and two factors are loaded under elements of Skill, Incentives and Resources respectively. Although these elements had

**Table 5. The five control factors critical to success of a BPI project.**

| Element Name | CSF | Conceptual basis Link to literature |
|---|---|---|
| **Vision** | Understanding of the Process | [57, 58, 59, 55] |
| | Value Realization | [61, 59, 60] |
| | Communication | [12, 13, 63, 62] |
| | Resource Allocation | [55, 56, 59, 57] |
| | Inv. Of organization Stakeholders and Leadership | [3, 63, 57, 61] |
| **Skills** | Appointment of Process Owners | [64, 67, 65, 63] |
| | People Change Management | [4, 67, 66] |
| **Incentive** | Performance Measurement | [68, 69] |
| **Resources** | Supporting Organizational Structure | [2, 76] |
| | People Training and empowerment | [71, 70] |
| **Action Plan** | Customer Focus | [13] |
| | Process Improvement Road Map | [74, 76] |
| | Scope change Management | [75, 72] |

two or less than two factors loaded, yet they were considered appropriate under the procedure carried out and furthermore the results are in-line with a number of previous researches as indicated in Table 5. This directs that for any BPI initiative the project actor must take account of the five change accelerators defined in terms of the industry relevant CSFs for the successful implementation of BPI projects.

**Vision.** Creating a shared vision that is deeply understood amongst all the team members is a hallmark for success of a BPI intervention [55]. Understanding of process is identified as an integral part of vision setting that transform processes from mobilization to implementation of change [56]. Some other researches have also established the link between understanding of process and vision [57, 58, 59]. It has been emphasized that when the value of the change initiatives are realized and aligned with vision, the change initiatives have a better chance of completion [60, 61]. Kotter introduced an eight step transformation process that impedes the transformation process [59]. Numerous other papers were found that established the importance of communicating vision and its impact on successful completion of improvement projects [62, 63, 64, 65].

Allocation of resources should be in alignment with the vision in order to lead the organization to the successful implementation of an improvement project [55, 56, 57, 60] Other studies have also emphasized the role of organizational stake holder and leadership in vision setting [55, 56, 58, 61].

**Skills.** It is evident from the literature that when process owners are equipped with the skills to manage the improvement project, the project has better chances of successful completion. It has been found that along with skill development for the tasks on job, organizations should develop managers and equip them with change management and implementation skills. These skills should be sought after both before and after their appointment as process owners of the BPI projects. With rapidly changing current demands, it is high time that organizations be proactive in developing the relevant skills [64, 65].

Management of change is an emerging area of study that is getting recognition widely in all sectors of businesses. People management through change requires ample skills. There are a number of frameworks that have emerged and organizations have spent a large amount of their budgets on consultants and trainers that educate the employees in this competitive

environment. Impact of skill development on the employees for people change management has been studied to facilitate successful completion of BPI projects [4,66, 67].

**Incentive.** It has been found that proper performance measurement and fair incentives positively influence in successful completion of the improvement initiative [68]. Performance measurement also serves as an incentive as it acknowledges the highly performing employees and also instills controls and checks on the employees [69].

**Resources.** Supporting organizational structure serves to be very crucial resource when it comes to implementation of an improvement project [70]. There have been numerous studies that discuss the impact of specific organizational structures and their impact on the BPI projects. It has been found that a flexible organizational structure is more supportive of the success of BPI projects [2].

Development of the employees on the undergoing improvement projects is very crucial for success. The employees should be trained and empowered to deliver the newly developed projects [71]. Employee training and empowerment is discussed in the literature in following two perspectives:

1. Training and empowerment on the proposed process

2. Training and empowerment over critical decision making and management of the change management process, impeding the successful completion of the BPI project [72].

**Action plan.** The contribution of a complete and concise roadmap in successful contribution of BPI projects is indubitable and well established. Customer focus is an integral part of all types of action plans [73]. Literature references suggest that process improvement roadmap is integral to develop a concrete action plan [74]. For an action plan to be comprehensive, it is imperative that it does not deviate from the scope. The evolution of business process management sheds light on the importance of scope change management for successful completions of the improvement projects [75, 76].

The extended model shown in Fig 3 thus contributes by drawing out the five keys agents of change- the "Change Accelerators" signifying their presence for successful implementation of a BPI Project. The extension defines the elements of change in context of the telecom industry BPI projects. Given realistic time and budget, project actor ensures project success, after taking into account these five key agents of change.

## Validity analysis

The following analysis were conducted in order to ensure the validity of the proposed model

**Content validity.** Validation of the model was done firstly, by showing the model and results to subject matter experts (BPMs)[53].

**Construct validity.** The performed EFA explains he contribution of the vision, skills, incentives, resources and action plan to the total variance explained thus ensuring the construct validity [53].

**External validity.** External validity establishes the generalizability of research to and across different times, settings and measures [77]. To establish external validity a t-test was performed [78]. An independent sample (sample 2) of about 50 was selected from the management of strategy department of two telecom organizations. The sample participants had experience of conducting BPI projects and were given the same questionnaire (used for EFA referred here as sample 1). The respondents were asked for their opinion about BPI project success factors; (5 most critical to success; 1 not critical at all). Similarly, 50 random responses (sample 1) were selected from the earlier EFA survey and an analysis was conducted using

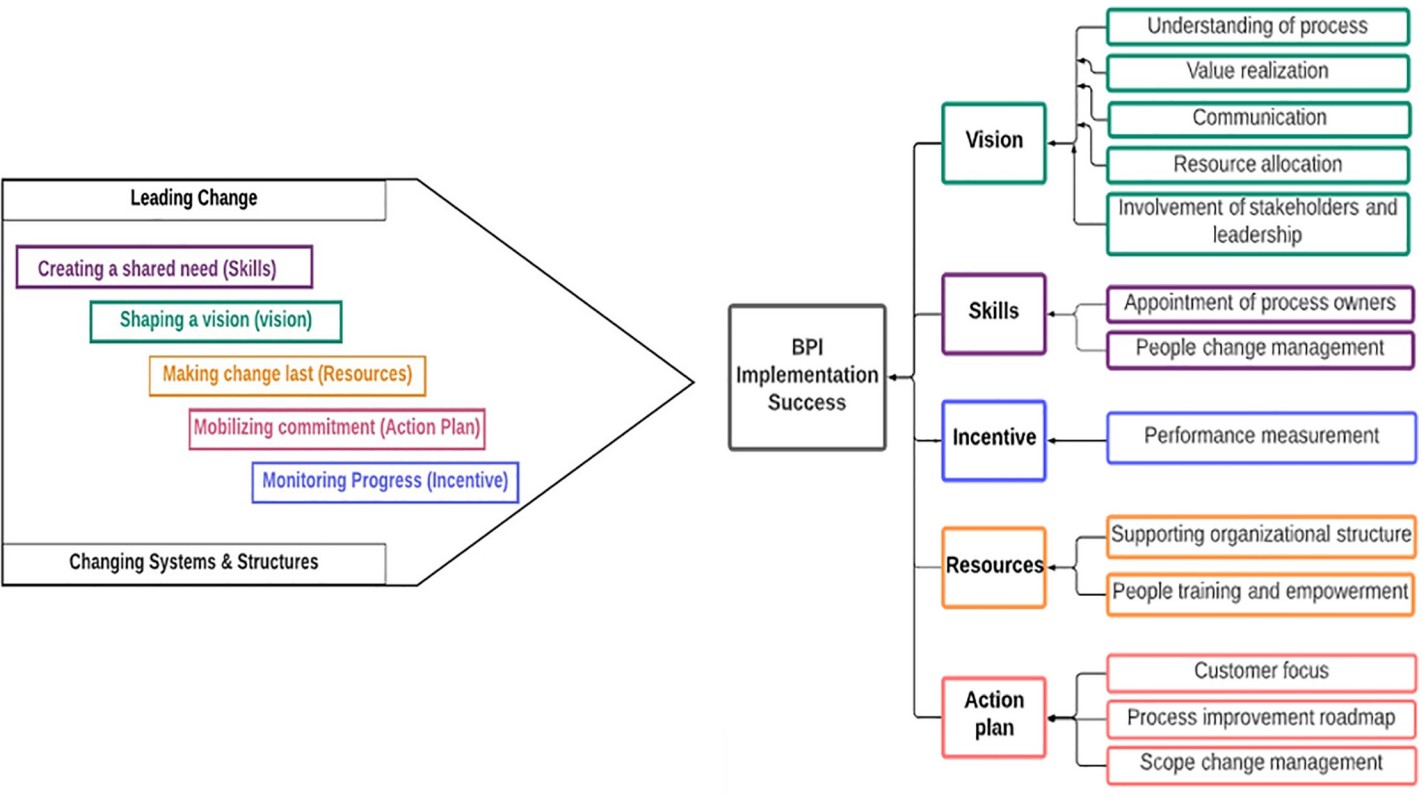

**Fig 3. Extension of the CAP model.**

independent sample t-test. The null hypothesis and the alternative hypothesis are:

$$If\ \mu_1\ and\ \mu_2\ are\ means\ of\ sample1\ and\ sample\ 2$$

$$H_{0:}\ \mu_1 = \mu_2$$

$$H_{1:}\ \mu_1 \neq \mu_2$$

Two tailed test with $\alpha$ = .05, df = 98 is applied for which the critical value is 1.9845. The results of the t-test are presented below in Table 6:

The group that participated in model design (M = 3.78, SD = 0.492) was not significantly different than the group that validated the model (M = 3.65, SD = 0.572), t (98) = 1.259, p = 0.211. As there is no significant difference among the two groups (people who participated in EFA, people who did not participate in EFA), hence we can conclude that the extension of the CAP model can be successfully used to ensure the success of BPI project.

**Table 6. Independent sample t-test for mean difference.**

| | | M | SD | t | p | 95% CI | | Cohen's d |
|---|---|---|---|---|---|---|---|---|
| | | | | | | LL | UL | |
| Success | ModelSuccess | 3.7800 | 0.4917 | 1.259 | 0.211 | -.0773 | 0.3458 | 0.254 |
| | ValidationSuccess | 3.6457 | 0.5716 | | | | | |

## Discussion

The purpose of this research was to design an extension of CAP model that ensures the success of BPI initiatives. To achieve this, the authors have first developed a thorough understanding of what factors impact the success of BPI projects. The study of literature has lead the authors to develop a list of CSFs that have been identified and tested in studies over last decades (attached as S1 Appendix). The authors then conducted a Delphi study with the involvement of business process managers that are experienced in BPI projects deemed as subject matter (BPI) experts. A consensus was derived from the experienced business process managers and was statistically tested using Delphi analysis.

During rounds of Delphi analysis the authors came across interesting observations. It is to be noted that when the ranks of the CSFs were calculated from the average score of each CSF [47] it was observed that there was a difference in opinion of the two groups. The "leadership" is the decision maker. This group decides why, when and how the improvement intervention takes place. The 'management' is the action actor as it makes the project happen. The difference in the point of view of the groups is evident as the leadership has a bird's eye view of the whole situation whilst the managers have the look and feel of the total picture. The differences in the point of view of the two groups are shown in detail in the radar diagram shown in Fig 4.

The derived most critical success factors were further analyzed using the exploratory factor analysis. EFA was used to explore the underlying theoretical structural details of the CSFs with respect to BPI project implementation success. The conduction of this analysis was a crucial and a challenging task with the involvement of BPI teams constituting 267 respondents from all the major telecom organizations. The responses generated interesting results.

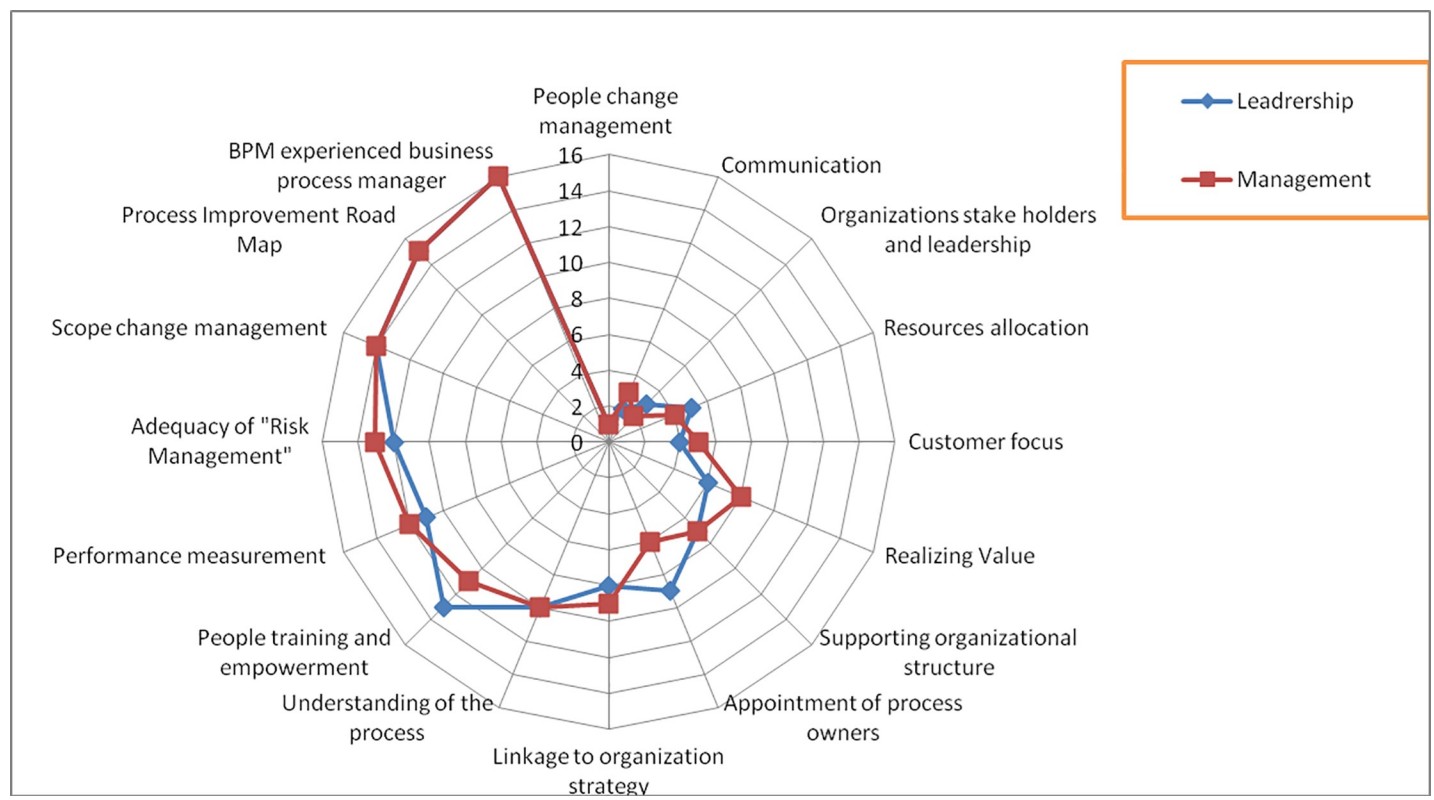

**Fig 4. Radar graph for the spread of opinions in round-2.**

Understanding of the process (with factor loading 0.865), value realization (0.834), communication (0.856), resource allocation (0.810) and involvement of stakeholders and leadership (0.765) showed most influence on the component 1. Similarly, appointment of process owners (0.802) and people change management (0.776) influenced component 2. While performance measurement (0.843) influenced component3 and supporting organizational structure (0.875), people training and empowerment (0.627) explained component 4. Scope change management (0.691), process improvement road map (0.548) and customer focus (-0.786) have most influence on component 5. It should be noted that for some components 2 or less than two factors have been loaded, this finding has been backed up by literature [79, 80, 81, 82], indicating that for some narrowly defined constructs, single-item measures may suffice. Another interesting finding is that customer focus is loaded onto the fifth component with a negative value which only represents the direction of the eigenvector and has no bearing on the interpretation of its magnitude. Customer focus had the highest loading which represents its importance relative to other elements in the factor. This finding is in compliance with latest research that directs that customer focus must be pragmatically addressed in BPI projects, and when not thoroughly looked after may hinder the successful implementation of BPI projects [83]. The authors thus needed to have a keen consideration on this while moving towards the next step.

Next step was to put under the microscope each identified component and establish an extension of the CAP model based on the five agents of change identified by CAP. The authors meticulously mapped each factor on the agents of change in light of the literature presented over the recent years. The paper contributes to the existing body of knowledge by presenting an extension of CAP for the BPI projects in the telecom industry of Pakistan. The extension presents a list of factors that must be taken into account for successful implementation of BPI projects.

The focus on the customer should not drive the BPI project away from the desired and intended results. Making use of the cardinal aspects of CAP for successful implementation is a distinctive approach in this area. The proposed CAP extension model is based upon the iterative and exploratory characteristics of Delphi technique used to meet research objectives. To understand the CSFs of BPI projects from the vast available success factors and to study their relevance to the telecom industry, the exploratory research method is considered appropriate [54]. The proposed model has twofold significance. It is prepared using critical factors from literature which were then ascertained by experienced experts from telecom industry. Moreover, the time tested CAP model integrated with BPI projects success is likely to improve chances of meeting the desired objectives.

Validity analysis was conducted on the model and content. Construct as well as the external validity has been ensured. The collaboration of literature and experience of industry personnel, who have practically implemented BPI projects and have endured the challenges is also a novel contribution to the existing body of knowledge.

## Utility of extended model

The proposed augmented model is designed with the help of telecom industry BPMs and teams making it applicable and relevant to this industry. Although the model has been designed by keeping telecom sector in view, it is of relevance to other related sectors as well where change management projects for process improvement are executed. The telecom industry is unique in terms of the technological advances and fierce competition it faces due to multiple external and internal factors. This extension directs BPM teams to ensure and expedite BPI project success. The biggest challenges in telecom sector relate to making change last and mobilizing commitment. The proposed model identifies critical factors to address these specific concerns.

## Conclusions

The extension of CAP for BPI success is a comprehensive tool that guides the business process manager with complete directions that ensure success of a BPI project. It directs that BPI should be made a part of the organization's vision. It defines the skill for BPI projects is the appointment of process owners and practicing people change management. The performance during the implementation of BPI projects should be measured for the successful implementation. Organizational structure that supports the improvement initiative and empowers employees are the key resource that contributes towards success. It is emphasized that improvement roadmap must be outlined with control over the scope changes, focusing on the customer needs.

The paper contributes by presenting industry relevant extension of CAP for BPI implementation that give the practitioners a crisp list of action items, when followed pave the way for successful implementation of BPI projects. The extension of CAP by defining the five key elements of change in context of BPI in Telecom industry, which is novel and unique.

## Limitations & future work direction

There are limitations of the research and some pose as future work opportunities. The research is limited to the telecom Industry of Pakistan which is a rapidly evolving industry, facing challenges on technological, political, economical, financial and human resource aspects. The adoption of the proposed extension in different industries facing similar challenges and for different geographical locations can be an interesting topic of a future study. Exploration of CSFs in other industries could be another area for future work. Exploring the relationship of customer focus and BPI action plans can be an interesting area of study. A case study analysis of the proposed model is also encouraged. This research directs practitioners by probing into the pitfalls of BPI implementation with a fresh perspective, encouraging them to make informed decisions. This is likely to expedite and enhance the chances of success of BPI initiatives.

## Supporting information

**S1 Appendix. List of critical factors from the literature along with references [84–94].**
(DOCX)

**S1 Dataset.**
(XLSX)

**S2 Dataset.**
(XLSX)

**S1 Questionnaires. Selection of most critical success factors.**
(DOCX)

## Author Contributions

**Conceptualization:** Maha Syed Ibrahim.

**Data curation:** Maha Syed Ibrahim.

**Formal analysis:** Maha Syed Ibrahim, Aamer Hanif.

**Investigation:** Maha Syed Ibrahim.

**Methodology:** Maha Syed Ibrahim, Aamer Hanif.

**Resources:** Maha Syed Ibrahim.

**Software:** Faheem Qaisar Jamal.

**Supervision:** Ali Ahsan.

**Validation:** Maha Syed Ibrahim, Aamer Hanif.

**Visualization:** Maha Syed Ibrahim, Faheem Qaisar Jamal.

**Writing – original draft:** Maha Syed Ibrahim.

**Writing – review & editing:** Maha Syed Ibrahim, Aamer Hanif.

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
