## [Decision Letter · Decision Letter 0]

25 Aug 2019

PONE-D-19-20062

Ensuring Business Process Improvement Success- An Extension of Change Acceleration Process Model

PLOS ONE

Dear Authors,

Thank you for submitting your manuscript to PLOS ONE. After careful consideration, we feel that it has merit but does not fully meet PLOS ONE’s publication criteria as it currently stands. Therefore, we invite you to submit a revised version of the manuscript that addresses the points raised during the review process.

Although the reviewers had a negative response to your article, however, you have a second chance for your article. the article will be revised after modification and this will be the final chance. so, I suggest you carefully follow the reviewers comments and respond in details for each one.

We would appreciate receiving your revised manuscript by 23 November 2019. To enhance the reproducibility of your results, we recommend that if applicable you deposit your laboratory protocols in protocols.io, where a protocol can be assigned its own identifier (DOI) such that it can be cited independently in the future. For instructions see: http://journals.plos.org/plosone/s/submission-guidelines#loc-laboratory-protocols

We look forward to receiving your revised manuscript.

Kind regards,

Amira M. Idrees, Associate Professor

Academic Editor

PLOS ONE

Journal Requirements:

2. Please include additional information regarding the survey or questionnaire used in the study and ensure that you have provided sufficient details that others could replicate the analyses. For instance, please provide a copy of the questionnaire you developed as part of this study in both the original language and English, as Supporting Information, unless it is under copyright, in which case, please provide a reference to the previous publication.

NO-The funders had no role in study design, data collection and analysis, decision to publish, or preparation of the manuscript.

a) Please provide an amended Funding Statement that declares *all* the funding or sources of support received during this specific study (whether external or internal to your organization) as detailed online in our guide for authors at http://journals.plos.org/plosone/s/submit-now.  

b) Please state what role the funders took in the study.  If any authors received a salary from any of your funders, please state which authors and which funder. If the funders had no role, please state: "The funders had no role in study design, data collection and analysis, decision to publish, or preparation of the manuscript."

Reviewers' comments:

Reviewer's Responses to Questions

**Comments to the Author**

1. Is the manuscript technically sound, and do the data support the conclusions?

Reviewer #1: No

Reviewer #2: Yes

2. Has the statistical analysis been performed appropriately and rigorously? 

Reviewer #1: I Don't Know

Reviewer #2: Yes

3. Have the authors made all data underlying the findings in their manuscript fully available?

Reviewer #1: No

Reviewer #2: No

4. Is the manuscript presented in an intelligible fashion and written in standard English?

Reviewer #1: No

Reviewer #2: Yes

5. Review Comments to the Author

Reviewer #1: Introduction and Literature review need to be improved.

The flow and structure of the paper is not well arranged.

Research design is prone to biases.

Based on the analysis, stating that the proposed model "ensure" BPI success is an extreme statement.

The novelty of this work is not clear. (Must be address in the discussion section)

The extended CAP model is not clearly defined. How is the proposed model better than existing methods?

Figures used are not clear. (instead of copy pasting images from the web, i suggest to duplicate it)

Outdated references must be updated.

Reviewer #2: Please refer to the attached report (document entitled 'PLOS ONE reviewer report PONE-D-19-20062') for the details of the report on this paper.

6. PLOS authors have the option to publish the peer review history of their article (what does this mean?). If published, this will include your full peer review and any attached files.

Reviewer #1: No

Reviewer #2: No

---

## [Author Response · Author response to Decision Letter 0]

24 Oct 2019

Responses have been uploaded in a separate file, and also below:

Response to reviewers

Concern1 Please ensure that your manuscript meets PLOS ONE's style requirements, including those for file naming

Response The authors have updated the style requirements along with file names

Concern2 Please include additional information regarding the survey or questionnaire used in the study and ensure that you have provided sufficient details that others could replicate the analyses

Response The survey questionnaires have been provided as supporting files

Concern3 The data should be provided as part of the manuscript or its supporting information, or deposited to a public repository.

Response The data files have been provided in supporting files

Concern4 While revising your submission, please upload your figure files to the Preflight Analysis and Conversion Engine (PACE) digital diagnostic tool, https://pacev2.apexcovantage.com/. PACE helps ensure that figures meet PLOS requirements.

Response Figure requirements have been checked.

Concern5 A dedicated (can also be brief if needed) section on the Pakistan telecommunication sector would also be good for readers to better appreciate the topic

Response The required section has been included in the manuscript in Page 2 as a subheading for introduction.

Concern6 Some brief background of the respondents may be good – e.g., age, position in organization. Ideally, these managers should be similar in terms of their job designation/title/position otherwise the responses collected may suffer from some position bias?

Response The brief background and demographics have been added under the sub-heading sample for both Delphi and EFA

Concern7 Did the authors consider adding any new CSFs? Why not? It seems that all the CSFs in the exhaustive list were sourced from other literatures. Did the questionnaire design involve any open-ended sections to elicit new information (potentially new CSF?)?

Response Delphi analysis invites participants to include any CSF they deem significant, in Round 1. in this case no additional CSF was suggested by the Panel.

Concern8 

Research design is prone to biases.

Response This point has been addresses in the section on “sampling and use of experts of Delphi” with a reference to literature.

Concern9 

Based on the analysis, stating that the proposed model "ensure" BPI success is an extreme statement.

Response The specified statement has been changed and also the paper title has been modified to address this extreme statement.

Concern10 

Outdated references must be updated.

Response The authors have updated the references and some new references have been added.

Concern11 Where’s Appendix B?

Response Reference to Appendix B was made by mistake and has been fixed.

Concern12 Some sort of alignment issue in column 1 in Table IV?

Response All the tables have been re-aligned and updated

Concern13 Figures used are not clear. (instead of copy pasting images from the web, i suggest to duplicate it)

Response All figures have been recreated.

Concern14 The authors should provide all the questionnaires in the Appendix (e.g., the Delphi questionnaires, the EFA questionnaire etc..). 

Response The questionnaires have been added in Appendix B

Concern15 In page 18 – perhaps a bit more explanation on why (and what does it mean) and the implications of customer focus having a negative factor loading

Response The implication of negative factor loading has been discussed in the paper by explaining it in the discussion section of the paper.

Concern16 Since there are many use of acronyms, the authors would be well-advised to provide the full term when the item appears for the first time – e.g., in page 2 first line of the second paragraph, what does GE stands for?

Response GE stands for General Electric. All the acronyms have been updated.

Concern17 While the paper is fairly well-written, it is not without its typos/grammatical issues and they are quite a few throughout the paper. Also, some parts appears to be a little sloppy (e.g., the word Figure does not need to be bolded, some in-text referencing and reporting of the sources of the figures [in the case of Fig 4 in page 4] not correct, etc..), spacing in between paragraphs a little neglected, etc… so some editorial improvements are necessary. In fact, I think it would be good to have the paper proofread (if possible) before further submission(s).

Response The paper has been proofread and updated accordingly.

Concern18 I am not sure if this is a concern but perhaps the authors can offer a more convincing discussion as to why the proposed extended model is so crucial, specifically to the telco industry – the explanation in the last paragraph of page 18 and first paragraph in page 19 offers very general discussions on the model and its applications.

Response The discussion has been improved to be clearer and convincing. A subsection “Utility of extended model” has been added to the paper.

Concern19 One may expect the discussions to be more industry specific or at the very least offer some insights as to how the model would contribute to making BPI projects more successful in the context of the telco sector.

Response The utility of extended model in telecom sector has been highlighted by adding an exclusive sub-section in the paper.

Concern20 The novelty of this work is not clear. (Must be address in the discussion section)

Response A subsection explaining the novelty of this work has been added.

Concern21 The extended CAP model is not clearly defined. How is the proposed model better than existing methods?

Response The rationale behind extended model is explained in subsection “Extension of CAP model”

Concern22 In page 7 (under Round 2 Ranking, Results & Analysis), the respondents were divided into 2 groups, i.e., 'Leadership' and 'Management'. How did the authors decide on the allocation of which respondent to which group? Also, how many were placed under Leadership and how many, Management? 

Response Director level and above have been grouped together as Leadership. 

Managers and senior managers have been grouped as 'Management'

Concern23 In page 8 (under Round 3 Ranking, Results & Analysis), how many respondents were there? 

Response 22 respondents, also reflected in the paper.

Concern24 In page 6, the number of respondents was 26 and 22 in the two rounds respectively – are these sufficient? The authors claimed that a previous literature indicated that the size may vary from 4 to 3000 --- does this suggests that the number of respondents in this paper is on the low side? Just a suggestion, perhaps to cite a previous work that has a sample size around this amount (e.g., anything around 30 respondents or less)? 

Response Yes the number of respondents is 22 which is applicable in this case. Respondents less than 15 have also been used in similar studies. Same has been reflected in the paper.

Concern25 Introduction and Literature review need to be improved. The flow and structure of the paper is not well arranged.

Response The manuscript has been revised and updated with improved structure.

Concern26 A section (or sub-section) devoted to BPI after the introduction section would improve the flow of the paper and also provide some general and basic insights to the use of BPI – this would be especially useful (easier) for general readers (non-specialists in the area).

Response The manuscript has been revised and a few lines devoted to BPI have been added

Concern27 The section on Change Accelerated Process Analysis should come after the proposed section on BPI and not after the Literature Review – this to improve the flow of the paper. 

Response The proposed changes have been made

Concern28 The literature review needs to be more critical and better informed – a little too brief given that it is less than a page long. 

Response The literature review has been revised

Concern29 The last paragraph in page 3 should not be part of the literature review – move this to the section on Research Design.

Response The proposed changes have been made in the paper.

---

## [Decision Letter · Decision Letter 1]

11 Nov 2019

Towards successful business process improvement – An extension of change acceleration process model

PONE-D-19-20062R1

Dear Author,

We are pleased to inform you that your manuscript has been judged scientifically suitable for publication and will be formally accepted for publication once it complies with all outstanding technical requirements.

With kind regards,

Amira M. Idrees, Associate Professor

Academic Editor

PLOS ONE

---

## [Editor Report · Acceptance letter]

14 Nov 2019

PONE-D-19-20062R1 

Towards successful business process improvement – An extension of change acceleration process model 

Dear Dr. Syed Ibrahim:

I am pleased to inform you that your manuscript has been deemed suitable for publication in PLOS ONE. Congratulations! Your manuscript is now with our production department. 

With kind regards,

on behalf of

Prof. Amira M. Idrees 

Academic Editor

PLOS ONE